# Analyzing cancer gene expression data through the lens of normal tissue-specificity

H. Robert Frost  *

Department of Biomedical Data Science, Geisel School of Medicine, Dartmouth College, Hanover, New Hampshire, United States of America

* rob.frost@dartmouth.edu

## Abstract

The genetic alterations that underlie cancer development are highly tissue-specific with the majority of driving alterations occurring in only a few cancer types and with alterations common to multiple cancer types often showing a tissue-specific functional impact. This tissue-specificity means that the biology of normal tissues carries important information regarding the pathophysiology of the associated cancers, information that can be leveraged to improve the power and accuracy of cancer genomic analyses. Research exploring the use of normal tissue data for the analysis of cancer genomics has primarily focused on the paired analysis of tumor and adjacent normal samples. Efforts to leverage the general characteristics of normal tissue for cancer analysis has received less attention with most investigations focusing on understanding the tissue-specific factors that lead to individual genomic alterations or dysregulated pathways within a single cancer type. To address this gap and support scenarios where adjacent normal tissue samples are not available, we explored the genome-wide association between the transcriptomes of 21 solid human cancers and their associated normal tissues as profiled in healthy individuals. While the average gene expression profiles of normal and cancerous tissue may appear distinct, with normal tissues more similar to other normal tissues than to the associated cancer types, when transformed into relative expression values, i.e., the ratio of expression in one tissue or cancer relative to the mean in other tissues or cancers, the close association between gene activity in normal tissues and related cancers is revealed. As we demonstrate through an analysis of tumor data from The Cancer Genome Atlas and normal tissue data from the Human Protein Atlas, this association between tissue-specific and cancer-specific expression values can be leveraged to improve the prognostic modeling of cancer, the comparative analysis of different cancer types, and the analysis of cancer and normal tissue pairs.

## Author summary

The frequency and functional impact of the genetic alterations that drive human cancer are highly tissue-specific. This tissue-specificity implies that important information about cancer biology can be extracted from the features of associated normal tissues. The use of normal tissue genomic data for cancer analysis has primarily focused on paired tumor

**Data Availability Statement:** The activity of human protein-coding genes in 21 common cancers and 18 associated normal tissues was determined using transcriptomic data from The Cancer Genome Atlas (TCGA) and transcriptomic data from the Human Protein Atlas (HPA). TCGA data

was accessed from the GDC Data Portal. TCGA PANCAN RNA-seq data was accessed from the file "GDC-PANCAN.htseq_fpkm.tsv.gz", which can be downloaded from https://gdc-hub.s3.us-east-1.amazonaws.com/download/GDC-PANCAN.htseq_fpkm.tsv.gz; TCGA phenotype data was accessed from the file "GDC-PANCAN.basic_phenotype.tsv", which can be downloaded from https://gdc-hub.s3.us-east-1.amazonaws.com/download/GDC-PANCAN.basic_phenotype.tsv.gz. For HPA, the HPA staff provided normal tissue gene expression data in the file "HPA.normal.FPKM.GDCpipeline.csv"; this data was specially normalized by the HPA group as FPKM using a pipeline similar to that employed by GDC for the TCGA data (this data was generated for the "Human Pathology Atlas" paper); this file is available at https://hrfrost.host.dartmouth.edu/CancerNormal/HPA.normal.FPKM.GDCpipeline.csv. Prognostic p-vlaues for each gene in each cancer type were retrieved from the HPA file "pathology.tsv.zip" downloaded from https://www.proteinatlas.org/download/pathology.tsv.zip. The Hallmark collection pathways were downloaded from version 7.0 of the Molecular Signatures Database (MSigDB) (as downloaded from http://software.broadinstitute.org/gsea/downloads.jsp). Gene set testing of the Hallmark pathways was performed using the pre-ranked version of the CAMERA method (the cameraPR() R function in the limma package).

**Funding:** Funding for HRF provided by National Institutes of Health grants K01LM012426, R21CA253408, P20GM130454 and P30CA023108. The funders had no role in study design, data collection and analysis, decision to publish, or preparation of the manuscript.

**Competing interests:** The authors have declared that no competing interests exist.

and adjacent normal samples. Less attention has been paid to pan-cancer approaches that use the general characteristics of normal tissue for cancer genomic analysis. To address this research gap, we explored the genome-wide association between the transcriptomes of 21 solid human cancers and their associated normal tissues as profiled in healthy individuals. We found a strong association between tissue-specific and cancer-specific expression, an association that can be leveraged to improve the prognostic modeling of cancer, the comparative analysis of different cancer types, and the analysis of cancer and normal tissue pairs.

## Introduction

The biology of human cancer is highly tissue and cell type-specific [1–5]. Most cancer driver genes are altered in only a small number of cancer types and, for drivers that are broadly mutated, the impact of the alteration often varies significantly between tissue types. Examples of genes with a tissue-specific pattern of alteration include von Hippel Lindau tumor suppressor (VHL) in renal cancer [6], adenomatous polyposis coli (APC) in colorectal cancer [7], and KRAS in pancreatic, lung and colorectal cancers [8]. Examples of driver genes that are altered in multiple cancer types but have a tissue-specific functional impact include BRCA1/BRCA2, which are impacted by germ-line mutation but lead to cancer primarily in estrogen-sensitive tissues (e.g., breast and ovaries) [9], and BRAF, which can be effectively inhibited in BRAF-mutated melanoma but not in BRAF-mutated colon cancer [10]. Only a small number of driver genes are broadly altered in many different cancers with similar functional consequences, e.g., TP53 [11] and MYC [12]. The tissue specificity of cancer driver genes is due to both cell intrinsic factors (i.e., epigenetic landscape and corresponding regulatory circuitry) and cell extrinsic factors (cell-cell interactions in the tissue microenvironment and environmental exposures). One important cell intrinsic factor is the cell type-specificity of proliferation drivers; 80–90% of the genes that drive proliferation function in only a limited number of cell types and these genes are frequently impacted by aneuploidy in cancer [2, 3]. Examples of cell extrinsic factors include exposure to estrogen in breast and ovarian tissue and the consequent vulnerability to BRCA1/BRCA2 mutations, and exposure to UV radiation in melanoma which leads to both increased sensitivity to the alteration of nucleotide excision repair genes and to an increase in the number of neoantigens and improved response to immunotherapy [4].

Given the significant tissue-specificity of cancer, the pattern of gene activity within normal tissue carries important information about the developmental trajectory of associated cancers and how those tumors will respond to therapeutic interventions. Efforts to leverage this information for the analysis of cancer genomics data have primarily focused on the analysis of paired tumor and adjacent normal tissue samples from repositories such as The Cancer Genome Atlas (TCGA) [13]. Analyses of paired tumor/normal data includes the work by Huang et al. exploring cancer prognosis prediction using data from adjacent normal samples [14] and research by Hu et al. investigating the pattern of relative gene expression between tumors and adjacent normal tissue [15]. Although the use of paired normal tissue data can improve the analysis of tumor data, this approach has two important limitations. First, adjacent normal tissue is impacted by the tumor microenvironment so is not accurate reflection of the biology of that tissue in a healthy individual. This so-called field cancerization effect explains why genomic data from adjacent normal tissue can independently predict cancer survival and, in some cases, provide superior performance relative to models based on tumor-

derived data [14]. The second limitation is the fact that data from adjacent normal tissue samples is often not available.

The limitations of paired tumor/normal data analysis motivate the exploration of associations between the general features of normal tissues, as measured in healthy individuals, and the characteristics of the cancer types that can develop in those tissues. While many researchers have investigated the association between normal tissue biology and cancer development in the context of specific cancer types and the associated cancer drivers (e.g., the association between estrogen sensitive tissues, BRCA1/BRCA2 mutations and cancer development [9, 16, 17]), only a limited number of researchers have explored the genome-wide association between normal tissue and cancer gene activity across multiple cancer types. The most prominent recent investigation into the general relationship between normal tissue and cancer gene activity is the work by the Uhlen et al. [18], who analyzed the association between gene expression in human solid tumors profiled by the TCGA and the corresponding normal tissues profiled by the Human Protein Atlas (HPA) [19]. Although the work by Uhlen et al. evaluated a large number of tumor types and normal tissues, the analysis focused on the association between mean gene expression in tumors and mean expression in the corresponding normal tissue, which revealed that the transcriptomes of normal tissues and cancers form two distinct clusters with each normal tissue more similar to other normal tissues than to the corresponding cancer type (see Uhlen et al. Fig S1). Successful use of normal tissue data for cancer genomics analysis was limited to liver cancer, where it was found that genes with elevated expression in normal liver tissue are favorably prognostic in liver cancer and genes whose expression was not specific to liver tissue are unfavorably prognostic. A similar liver-specific analysis was conducted by Li et al. [20], which revealed a concordant finding regarding the prognostic power of liver-specific genes.

To address the gap in effective pan-cancer approaches for leveraging normal tissue gene activity for the analysis of cancer genomics data, we explored the genome-wide association between the transcriptomes of 21 solid human cancers and their associated normal tissues as profiled in healthy individuals. While the average gene expression profiles of normal and cancerous tissue may appear distinct, as found by Uhlen et al. [18] with normal tissues more similar to other normal tissues than to the associated cancer types, when transformed into relative expression values, i.e., the ratio of expression in one tissue or cancer relative to the mean in other tissues or cancers, the close association between gene expression in normal tissues and related cancers is revealed. As we demonstrate through an analysis of tumor data from TCGA and normal tissue data from the HPA, this association between tissue-specific and cancer-specific expression values can be leveraged to improve the prognostic modeling of cancer, the comparative analysis of different cancer types, and the analysis of cancer and normal tissue pairs.

## Materials and methods

The results presented in this paper are based on bulk RNA-seq data from TCGA for 21 human solid cancers and from the HPA for the associated 18 normal human tissues (see Table 1 below for a list of the cancer types and matching normal tissues; additional information about these TCGA cancer types can be found in Table A in S1 Text). These cancer types were selected based on the availability of gene expression data for the corresponding normal tissues in the HPA. A similar set of 17 TCGA cancer types and paired normal tissues were selected for the analysis by Uhlen et al. [18]. In contrast to Uhlen et al., we have separately analyzed the three renal cancer types (kidney chromophobe, kidney renal clear cell carcinoma, and kidney renal

**Table 1. The 21 analyzed TCGA cancer types and corresponding HPA normal tissues.** The 'Cancer/normal correlation' column contains the Spearman rank correlation between the mean gene expression values in the TCGA cancer samples (i.e., the $c_{i,j}$ statistics) and the mean expression values in the associated HPA normal tissue (i.e., the $n_{i,j}$ statistics). The 'Most correlated' column lists the HPA normal tissue whose mean expression had the highest rank correlation with cancer mean expression. A value of '*' indicates that the associated normal tissue had the largest correlation.

| TCGA abbrev. | Cancer type | HPA tissue | Cancer/normal correlation | Most correlated |
|---|---|---|---|---|
| BLCA | Bladder Urothelial Carcinoma | urinary bladder | 0.901 | * |
| BRCA | Breast Invasive Carcinoma | breast | 0.924 | * |
| CESC | Cervical Squamous Cell Carcinoma and . . . | cervix, uterine | 0.852 | urinary bladder |
| COAD | Colon Adenocarcinoma | colon | 0.93 | * |
| GBM | Glioblastoma Multiforme | cerebral cortex | 0.895 | * |
| HNSC | Head and Neck Squamous Cell Carcinoma | tonsil | 0.862 | skin |
| KICH | Kidney Chromophobe | kidney | 0.932 | * |
| KIRC | Kidney Renal Clear Cell Carcinoma | kidney | 0.931 | * |
| KIRP | Kidney Renal Papillary Cell Carcinoma | kidney | 0.925 | * |
| LIHC | Liver Hepatocellular Carcinoma | liver | 0.929 | * |
| LUAD | Lung Adenocarcinoma | lung | 0.925 | * |
| LUSC | Lung Squamous Cell Carcinoma | lung | 0.886 | urinary bladder |
| OV | Ovarian Serous Cystadenocarcinoma | ovary | 0.817 | cervix, uterine |
| PAAD | Pancreatic Adenocarcinoma | pancreas | 0.886 | stomach |
| PRAD | Prostate Adenocarcinoma | prostate | 0.951 | * |
| READ | Rectum Adenocarcinoma | rectum | 0.912 | colon |
| SKCM | Skin Cutaneous Melanoma | skin | 0.838 | urinary bladder |
| STAD | Stomach Adenocarcinoma | stomach | 0.921 | * |
| TGCT | Testicular Germ Cell Tumors | testis | 0.698 | urinary bladder |
| THCA | Thyroid Carcinoma | thyroid gland | 0.934 | * |
| UCEC | Uterine Corpus Endometrial Carcinoma | endometrium | 0.868 | cervix, uterine |

papillary cell carcinoma), separately analyzed colon cancer and rectal cancer, and separately analyzed lung adenocarcinoma and lung squamous cell carcinoma.

TCGA and HPA RNA-seq data, normalized as FPKM+1 values, were used to compute several statistics that capture different aspects of gene activity and prognostic value. So that readers can more easily follow the main results presented below, a concise definition of these statistics is included in Table 2. More detailed information regarding the data sources, computation of the statistics in Table 2 and generation of other tables and figures can be found in S1 Text. A companion website for this paper (https://hrfrost.host.dartmouth.edu/CancerNormal/) provides access to files that hold the statistics in Table 2 for all analyzed normal tissues and cancers as well as a specially processed version of the HPA RNA-seq data.

## Results and discussion

### Association between gene activity in normal and neoplastic tissue

Fig 1 illustrates the projection of the 21 analyzed cancer types and 18 corresponding normal tissue types onto the first two principal components (PCs) from a principal component analysis (PCA) of mean expression values, i.e., PCA of a matrix that contains all $c_{i,j}$ and $n_{i,j}$ statistics. A scree plot of PC variances and projections onto PCs 3–8 are included as Figs A and B in S1 Text. As seen in Fig S1 from the Ulhen et al. [18] paper, cancers and normal tissues cluster separately in the space of the first 2 PCs according to gene expression data (i.e., cancers are more similar to each other than to the corresponding normal tissues), with the separation primarily driven by PC 1. Liver cancer and normal liver tissue are noticeable outliers in this projection

**Table 2. Gene-level statistics computed on TCGA and HPA RNA-seq gene expression data and TCGA survival data.**

| Statistic | Description |
|---|---|
| $c_{i,j}$ | Mean expression of gene $i$ in cancer type $j$. The average of these mean expression values for gene $i$ across all 21 cancer types is represented by $\bar{c}_i$. Expression values are normalized as FPKM+1. |
| $n_{i,j}$ | Mean expression of gene $i$ in normal tissue $j$ (the tissue associated with cancer type $j$). The average of these mean expression values for gene $i$ across all 18 normal tissues is represented by $\bar{n}_i$. Expression values are normalized as FPKM+1. |
| $c^*_{i,j} = \log_2(c_{i,j}/\bar{c}_i)$ | Cancer-specific expression of gene $i$ in cancer type $j$ as computed by the log fold-change in expression of gene $i$ between cancer type $j$ and the average in all cancer types. Large positive values represent genes whose expression in cancer $j$ is much larger than the average found in all profiled cancers, large negative values represent genes whose expression in cancer $j$ is much smaller than the average found in other cancers and values close to 0 represent genes whose expression is similar to that found in other cancers. |
| $n^*_{i,j} = \log_2(n_{i,j}/\bar{n}_i)$ | Normal tissue-specific expression of gene $i$ in tissue $j$ as computed by the log fold-change in expression of gene $i$ between tissue $j$ and the average in all normal tissues. The interpretation of these statistics is similar to that outlined above for the $c^*_{i,j}$ statistics. |
| $r_{i,j} = \log_2(c_{i,j}/n_{i,j})$ | Relative expression of gene $i$ between cancer type $j$ and the corresponding normal tissue as computed by the log fold-change in mean expression of gene $i$ in cancer $j$ and normal tissue $j$. |
| $s_{i,j}$ | Cancer prognostic value of gene $i$ for cancer type $j$ as computed by the signed log of a p-value generated by Uhlen et al. [18] to capture the significance of the association between expression of gene $i$ and survival for cancer type $j$. For genes with a favorable prognostic value, the -log of the p-value is used to produce positive values. For genes with an unfavorable prognostic value, the log of the p-value is used instead to generate negative values. |

and, unlike most other profiled cancer/normal pairs, have mean transcriptomes that are more similar to each other than to other cancers or normal tissues. This strong association between normal liver and liver cancer gene expression was identified and explored by both Uhlen et al. [18] and Li et al. [20]. Despite the separation between cancer and normal tissue transcriptomes

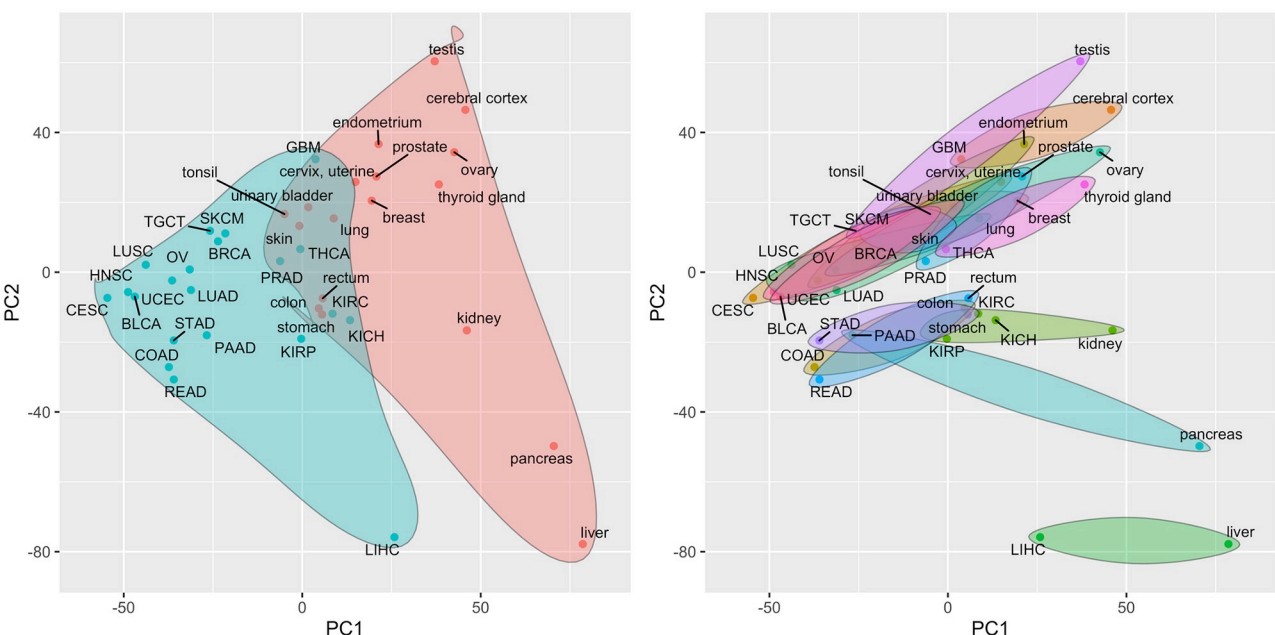

**Fig 1. Projection of TCGA cancer types and associated HPA normal tissues onto the first two principal components of the mean expression values.** Principal components are computed from a matrix of mean gene expression values, i.e., a matrix containing all $c_{i,j}$ and $n_{i,j}$ statistics. a) Cancers are represented by blue points and are enclosed in the blue shaded region; normal tissues are represented by red points and are enclosed in the red shaded region. b) Each normal tissue and the associated cancer type(s) are enclosed in a separate shaded region.

**Table 3. Distances between each cancer type and the corresponding normal tissue in reduced principal component (PC) space.** The Euclidean distance was computed between the projections of each cancer type and the associated normal tissue in the space spanned by the first two PCs of the mean gene expression matrix (as shown in Figs 1 and 2). The 'Cancer/normal relative distance' column contains the ratio of the distance between each cancer and normal tissue pair as visualized in Fig 1 to the average distance between the cancer and all other cancers or normal tissues. The 'Cancer/normal-specific relative distance' column contains a similar distance ratio computed in the space spanned by the first two PCs of cancer-specific and normal tissue-specific mean expression values as visualized in Fig 2. The 'Distance ratio' column contains the ratio of the 'Cancer/normal-specific relative distance' to the 'Cancer/normal relative distance'.

| TCGA abbrev. | HPA tissue | Cancer/normal relative distance | Cancer/normal-specific relative distance | Distance ratio |
|---|---|---|---|---|
| BLCA | urinary bladder | 0.997 | 0.122 | 0.123 |
| BRCA | breast | 1.06 | 0.455 | 0.43 |
| CESC | cervix, uterine | 1.25 | 0.576 | 0.461 |
| COAD | colon | 0.805 | 0.256 | 0.318 |
| GBM | cerebral cortex | 0.922 | 0.184 | 0.199 |
| HNSC | tonsil | 0.871 | 0.243 | 0.279 |
| KICH | kidney | 0.749 | 0.46 | 0.614 |
| KIRC | kidney | 0.916 | 0.574 | 0.627 |
| KIRP | kidney | 1.07 | 0.51 | 0.475 |
| LIHC | liver | 0.585 | 0.37 | 0.632 |
| LUAD | lung | 1 | 0.168 | 0.168 |
| LUSC | lung | 1.03 | 0.444 | 0.431 |
| OV | ovary | 1.81 | 0.584 | 0.322 |
| PAAD | pancreas | 2.18 | 2.39 | 1.1 |
| PRAD | prostate | 0.964 | 0.313 | 0.324 |
| READ | rectum | 0.829 | 0.311 | 0.375 |
| SKCM | skin | 0.497 | 1.04 | 2.1 |
| STAD | stomach | 0.818 | 0.435 | 0.532 |
| TGCT | testis | 1.81 | 0.857 | 0.474 |
| THCA | thyroid gland | 1.14 | 0.216 | 0.189 |
| UCEC | endometrium | 1.47 | 0.245 | 0.167 |

along the first PC, a very high correlation still exists between the mean gene expression values in a specific cancer and the expression values in the corresponding normal tissue. This association is captured in Table 1, which lists the rank correlation between cancer and associated normal tissue mean expression values, i.e., the $c_{i,j}$ and $n_{i,j}$ statistics. As shown in Table 1, the majority of the profiled cancers are most strongly correlated with their corresponding normal tissue. The close association cancer and normal tissue gene expression shown in Table 1 is consistent with the projections onto lower variance PCs shown in Fig B in S1 Text; among the first 8 PCs, only PCs 1 and 3 show a clear separation between cancers and normal tissues.

## Association between normal tissue-specificity and cancer type-specificity

When analyzing tissue-specific or cancer-specific gene expression, i.e., the ratio of expression in one tissue or cancer to the mean expression in all tissues or cancers as quantified by the $c_{i,j}^*$ and $n_{i,j}^*$ statistics, cancers and normal tissues no longer cluster separately and most cancers look very similar to the corresponding normal tissue (Tables 1 and 3 provide a quantification of this similarity). Fig 2 illustrates the projection of the analyzed cancers and normal tissues onto the first two PCs of the relative mean expression values, i.e., the PCs of a matrix combining all $c_{i,j}^*$ and $n_{i,j}^*$ statistics. Similar projects onto PCs 3–8 of the relative mean expression values can be found in Fig C in S1 Text. As seen in Fig 2, cancers and normal tissues no longer separate along PC 1 and each cancer tends to be very close to the corresponding normal tissue.

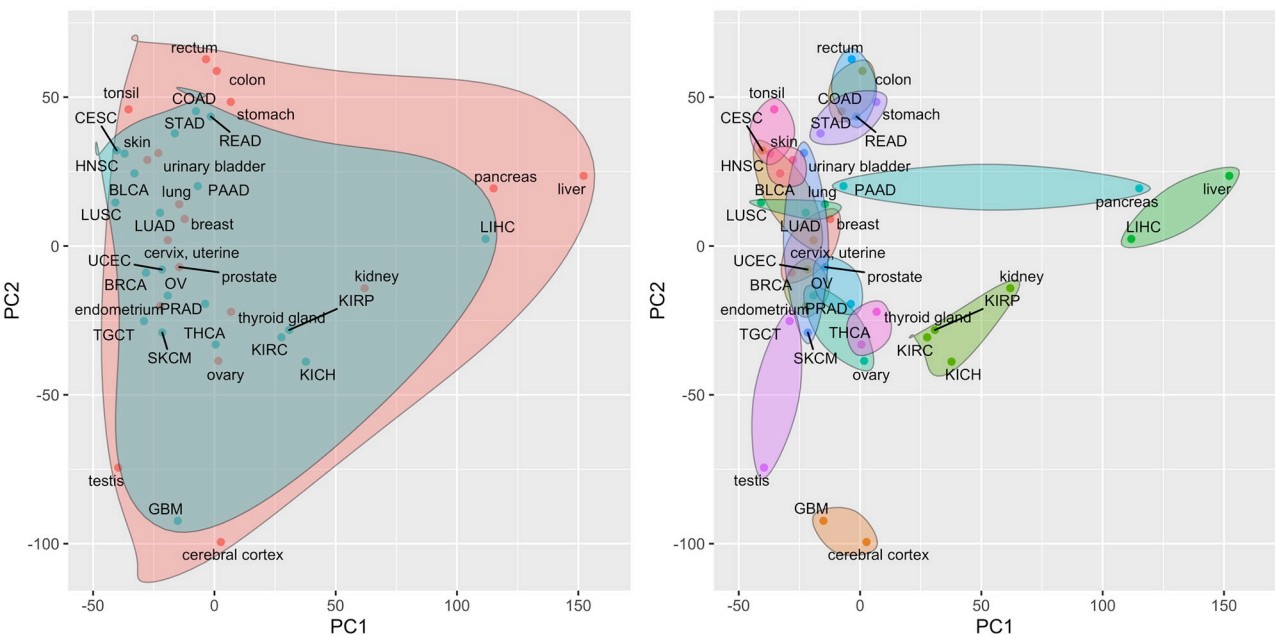

**Fig 2. Projection of TCGA cancer types and associated HPA normal tissues onto the first two principal components of the tissue/cancer-specific mean expression values.** Principal components are computed from a matrix of tissue-specific and cancer-specific mean gene expression values, i.e., a matrix containing all $c_{i,j}^*$ and $n_{i,j}^*$ statistics. a) Cancers are represented by blue points and are enclosed in the blue shaded region; normal tissues are represented by red points and are enclosed in the red shaded region. b) Each normal tissue and the associated cancer type(s) are enclosed in a separate shaded region.

These findings are quantified in Table 3 using the Euclidean distances between cancer and normal tissues in the space of the first two PCs. Specifically, the 'Cancer/normal relative distance' column contains the ratio of the distance between a given cancer and normal tissue in the PC space visualized by Fig 1 relative to average distance between that cancer and all other cancers or normal tissues. A relative distance of 1.0 indicates that the cancer mean expression values are no more similar to the expression values in the corresponding normal tissue than to a randomly selected cancer or normal tissue. Values above 1.0 indicate less similarity than would be expected at random and values below 1.0 indicate that the cancer and normal tissue transcriptomes are more similar than would be expected at random. The 'Cancer/normal-specific relative distance' column contains equivalent relative distance values generated in the space of the first two PCs computed on the cancer and tissue-specific expression data as visualized in Fig 2. The 'Distance ratio' column contains the ratio of the cancer/tissue-specific relative distances to the non-specific relative distances. Importantly, the distance ratios for most cancer/normal pairs are significantly below 1.0, i.e., cancer and normal tissue transcriptomic profiles are much more similar when evaluated in terms of cancer/tissue-specific values. Pancreatic cancer and skin cancer are two notable exceptions. For skin cancer, this result is likely due to the fact that the TCGA expression data is generated on metastatic lesions rather than primary tumors, which will reflect the properties of the host tissue. The implication of these results is that differences in gene expression between cancers largely reflect differences found between the corresponding normal tissues. A version of Table 3 with Euclidean distances computed on all PCs with non-zero variance can be found as Table B in S1 Text. While the distance between cancers and associated normal tissues in the space of all PCs is smaller for most cancer types when using tissue/cancer-specific expression values as compared to unadjusted values, the impact is much less pronounced relative to the distances computed on just PCs 1 and 2.

This is consistent with the concentration of cancer/normal transcriptomic differences in PCs 1 and 3.

## Association between normal tissue-specificity, cancer/normal differential expression and cancer survival

The ratio of cancer to normal tissue gene expression is inversely associated with normal tissue specificity for most of human genes in all tested cancer/normal pairs. In other words, tissue-specific genes tend to be down-regulated in cancer, which is consistent with a view of cancer as a dedifferentiation process [21]. This finding is quantified by the $\rho_{case/ctrl}$ values in Table 4, which capture the rank correlation between normal tissue specificity ($n_{i,j}^*$) and cancer/normal relative expression ($r_{i,j}$) and are negative for all 21 profiled cancer types. A related finding is that normal tissue specificity tends to be positively associated with favorable survival in cancer, i.e., an increase in expression of normal tissue specific genes is associated with improved survival in cancer. Genes that are not tissue-specific have the inverse association, i.e., an increase in expression is associated with worse survival. Genes that are down-regulated in a tissue relative to other tissues tend to have no survival association. The rationale behind these survival associations is that cancer cells are more dedifferentiated than normal tissue cells [18]. This survival association is quantified by the $\rho_{surv}$ values in Table 4, which capture the rank correlation between normal tissue specificity ($n_{i,j}^*$) and prognostic ability ($s_{i,j}$) and are near zero or

**Table 4. Correlations between normal tissue-specific gene weights and either cancer/normal relative expression or an indicator of cancer prognostic ability.** The $\rho_{cancer/norm}$ column holds the Spearman rank correlation between normal tissue-specific gene weights ($n_{i,j}^*$) and the log ratio of mean expression in the cancer type to mean expression in the normal tissue ($r_{i,j}$). The $\rho_{surv}$ column holds the Spearman rank correlation between normal tissue-specific gene weights ($n_{i,j}^*$) and the signed log of the p-value from a Kaplan-Meir test of the association between gene expression and cancer survival as computed by Uhlen et al. [18]) ($s_{i,j}$, which is computed as -log(p-value) for favorable genes and log(p-value) for unfavorable genes).

| TCGA abbrev. | HPA tissue | $\rho_{cancer/norm}$ | $\rho_{surv}$ |
|---|---|---|---|
| BLCA | urinary bladder | -0.299 | -0.0229 |
| BRCA | breast | -0.437 | -0.0993 |
| CESC | cervix, uterine | -0.432 | -0.105 |
| COAD | colon | -0.057 | 0.304 |
| GBM | cerebral cortex | -0.446 | 0.0764 |
| HNSC | tonsil | -0.48 | 0.11 |
| KICH | kidney | -0.232 | 0.454 |
| KIRC | kidney | -0.436 | 0.454 |
| KIRP | kidney | -0.311 | 0.454 |
| LIHC | liver | -0.358 | 0.35 |
| LUAD | lung | -0.363 | 0.073 |
| LUSC | lung | -0.406 | 0.073 |
| OV | ovary | -0.546 | -0.0348 |
| PAAD | pancreas | -0.643 | 0.148 |
| PRAD | prostate | -0.179 | -0.0397 |
| READ | rectum | -0.208 | 0.349 |
| SKCM | skin | -0.444 | -0.0707 |
| STAD | stomach | -0.258 | 0.344 |
| TGCT | testis | -0.636 | 0.256 |
| THCA | thyroid gland | -0.441 | -0.0851 |
| UCEC | endometrium | -0.45 | 0.0276 |

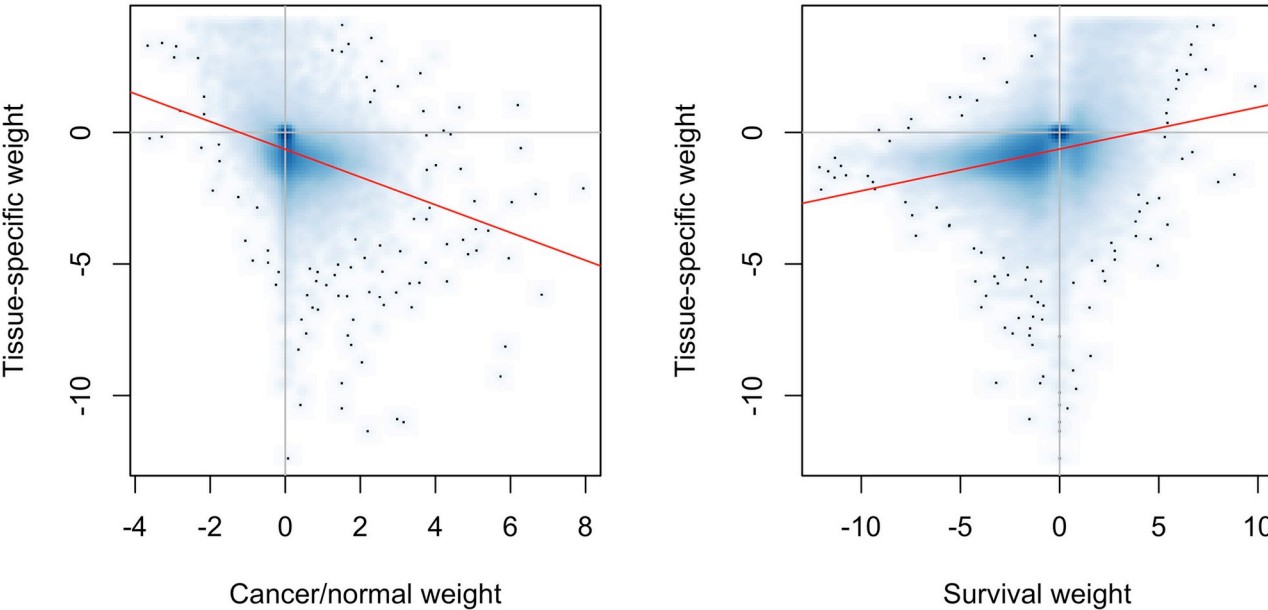

**Fig 3. Association of liver-specificity ($n^*_{i,\text{liver}}$), liver cancer/normal liver differential expression ($r_{i,\text{LIHC}}$) and liver cancer prognostic value ($s_{i,\text{LIHC}}$).** Each point in both plots represents a single gene and the red lines reflect the linear regression fit. a) Association between liver-specificity and differential expression between liver cancer and normal liver. b) Association between liver-specificity and prognostic value of gene expression for liver cancer.

positive for all 21 profiled cancer types. Fig 3 provides a more detailed visualization of the association of normal tissue-specificity with cancer/normal relative expression and cancer survival for liver cancer. Similar plots for the other profiled cancer types can be found in Figs D and E in S1 Text.

## Using normal tissue gene activity to improve the comparative analysis of cancers

As illustrated in Fig 2 and Table 3, the relative expression of genes between different cancer types is associated with the relative expression in the corresponding normal tissues. This association can be leveraged to improve the comparative analysis of cancer types based on transcriptomic data. Specifically, when attempting to identify genes that are differentially expressed (DE) between two cancer types, genes that are DE between the associated normal tissues can be prioritized, e.g., via gene filtering or a hypothesis weighting scheme like weighted false discovery rate (wFDR) [22]. Although this will improve the power for detecting DE genes between cancer types, the results will highlight differences between the underlying normal tissues rather than differences between the cancer types that are independent of normal tissue physiology. Alternatively, removing tissue-specific genes (or gene sets) can enable the identification of differences between cancer types that are due to malignant processes and not differential gene activity within the normal tissues. This second approach is one we have successfully used in the past for the analysis of gene expression in primary colorectal tumors and colorectal metastatic lesions in the liver and lung [23]. The pan cancer impact of this approach is illustrated in Fig 4 and Table 5. Fig 4 visualizes the Pearson correlation between the fold-change in gene expression between each pair of cancers and the corresponding pair of normal tissues. To compute this correlation for cancers $a$ and $b$, we first create two vectors of gene expression fold-change values: $\mathbf{v_c} = \{c_{1,a}/c_{1,b},\ldots,c_{p,a}/c_{p,b}\}$ and $\mathbf{v_n} = \{n_{1,a}/n_{1,b},\ldots,n_{p,a}/n_{p,b}\}$, where $\mathbf{v_c}$ holds the expression fold-change values for all $p$ measured genes between cancers $a$

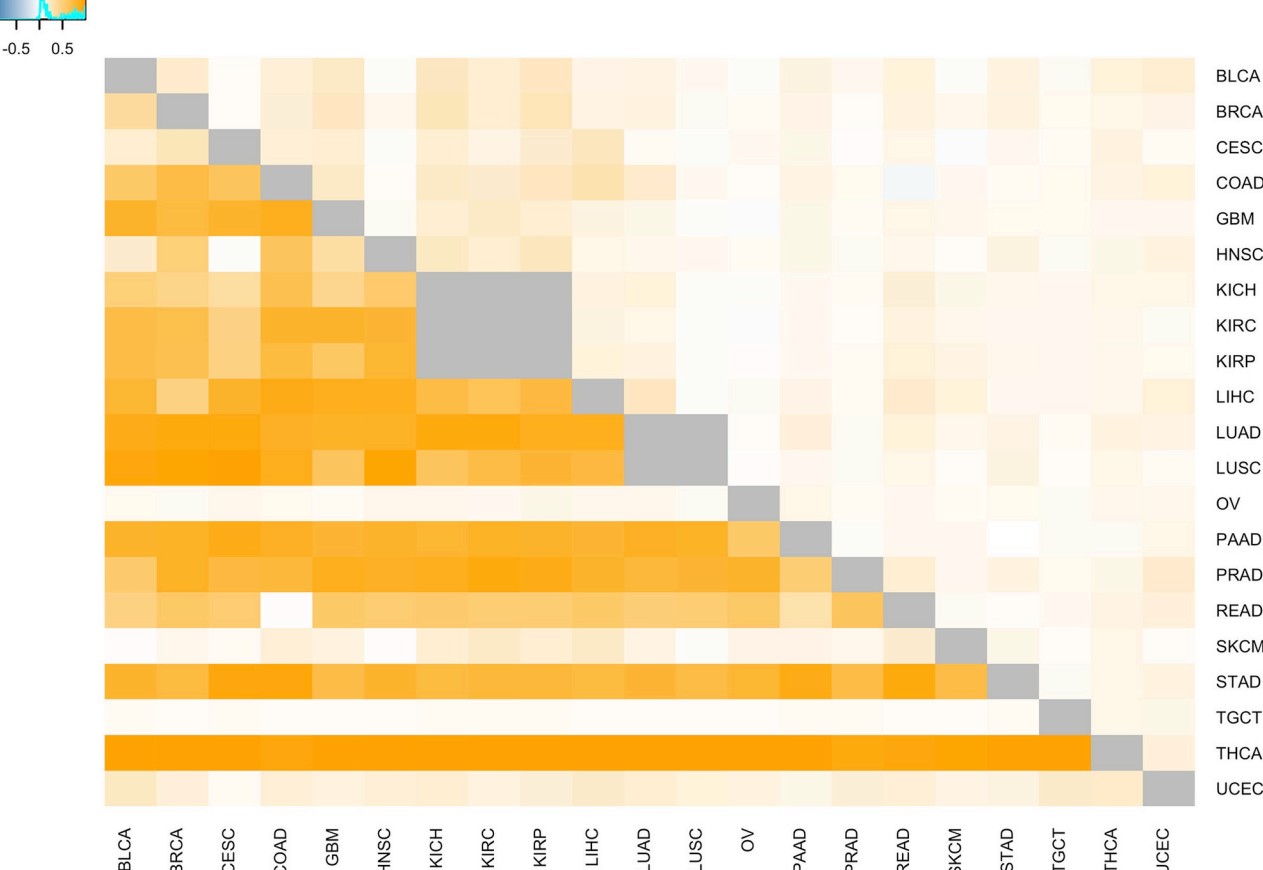

**Fig 4. Impact of normal tissue-specificity on cancer differential expression.** Cell coloring reflects the Pearson correlation between the fold-change in gene expression between each pair of cancers and the corresponding pair of normal tissues. Large correlations indicate that the expression difference between cancers mirrors the expression difference in the normal tissues. For cells below the diagonal, all genes are used to compute the correlations; for cells above the diagonal, genes that have a fold-change between normal tissues of ≤0.5 or ≥2 are excluded. As indicated by the lower correlations in cells above the diagonal, filtering out genes that show large differential expression between normal tissues will generate cancer differential expression results that are distinct from the normal tissue biology.

and $b$ and $\mathbf{v_n}$ holds the expression fold-change values between the normal tissues associated with cancers $a$ and $b$. The Pearson correlation coefficient is then computed between vectors $\mathbf{v_c}$ and $\mathbf{v_n}$ and these correlation coefficients are visualized in the cells below the diagonal in Fig 4. For cells above the diagonal, genes that have a fold-change between normal tissues (i.e., $n_{i,a}/n_{i,b}$) of ≤0.5 or ≥2 are excluded from the $\mathbf{v_c}$ and $\mathbf{v_n}$ vectors before computing the correlation. The positive correlation values generated on all genes indicate that the pattern of DE between most cancers mirrors the pattern of DE found between the corresponding normal tissues. The lower correlation values in the cells above the diagonal confirms that filtering out genes showing large DE between associated normal tissues will generate cancer DE results that are distinct from the normal tissue biology. Table 5 demonstrates the impact of this association for the comparative analysis of liver cancer and glioblastoma. This table lists the 10 Molecular Signatures Database (MSigDB) [24] Hallmark pathways most significantly associated with relative gene expression between liver and cerebral cortex and between liver cancer and glioblastoma according to the pre-ranked version of the CAMERA gene set testing method [25]. Gene set testing of liver cancer/glioblastoma relative expression was performed using all genes (middle

**Table 5. Impact of normal tissue-specificity on comparative analysis of liver cancer and glioblastoma.** The table lists the 10 MSigDB Hallmark pathways most significantly associated with relative gene expression between liver and cerebral cortex and between liver cancer and glioblastoma. Gene set testing of cancer relative expression was performed using all genes (middle column) and using genes filtered according to normal tissue differential expression (right column). Filtering specifically removed all genes whose log2 fold-change in expression between liver and cerebral cortex was $\leq -2$ or $\geq 2$. Without filtering, differential gene expression between liver cancer and glioblastoma is similar to differential expression between normal liver and cerebral cortex.

| rank | liver vs. cerebral cortex | liver cancer vs. glioblastoma | liver cancer vs. glioblastoma (filtered) |
|---|---|---|---|
| 1 | XENOBIOTIC_METABOLISM | XENOBIOTIC_METABOLISM | E2F_TARGETS |
| 2 | FATTY_ACID_METABOLISM | FATTY_ACID_METABOLISM | G2M_CHECKPOINT |
| 3 | BILE_ACID_METABOLISM | BILE_ACID_METABOLISM | EPITHELIAL_MESENCHYMAL_TR... |
| 4 | COAGULATION | OXIDATIVE_PHOSPHORYLATION | MYC_TARGETS_V1 |
| 5 | INTERFERON_GAMMA_RESPONSE | ADIPOGENESIS | KRAS_SIGNALING_UP |
| 6 | TNFA_SIGNALING_VIA_NFKB | PEROXISOME | INFLAMMATORY_RESPONSE |
| 7 | ALLOGRAFT_REJECTION | COAGULATION | TNFA_SIGNALING_VIA_NFKB |
| 8 | IL6_JAK_STAT3_SIGNALING | CHOLESTEROL_HOMEOSTASIS | IL6_JAK_STAT3_SIGNALING |
| 9 | ADIPOGENESIS | REACTIVE_OXYGEN_SPECIES_P... | TGF_BETA_SIGNALING |
| 10 | INFLAMMATORY_RESPONSE | KRAS_SIGNALING_DN | P53_PATHWAY |

column) and using genes filtered according to DE between normal liver and cerebral cortex (right column). Filtering for this analysis removed all genes whose fold-change in expression between liver and cerebral cortex was less than $\leq 0.5$ or $\geq 2$. As seen in the table, differential pathway activity between liver cancer and glioblastoma is similar to differential pathway activity between normal liver and cerebral cortex when all genes are considered. When genes exhibiting significant normal tissue DE are removed, the comparative cancer results are distinct from the normal tissue results.

## Use of normal tissue-specificity to improve the comparative analysis of normal and neoplastic tissue

As illustrated in Table 4, tissue-specific genes are more likely to be down-regulated in cancer as compared to the associated normal tissue and non-specific genes are more likely to be up-regulated; these associations hold across all 21 evaluated cancer types. Similar to the approach outlined in Section Using normal tissue gene activity to improve the comparative analysis of cancers for the comparative analysis of different cancers, this association between normal tissue-specificity and cancer/normal DE can be used to prioritize genes for analysis via either gene filtering or hypothesis weighting. Fig 5 illustrates the increase in power that can be achieved by filtering genes according to normal tissue-specificity prior to a cancer/normal DE analysis. Specifically, this figure contains quantile-quantile (Q-Q) plots of p-values from gene set testing of MSigDB Hallmark pathways relative to cancer/normal relative expression (i.e, the $r_{i,j}$ statistics) for the 21 profiled cancer types using all genes or genes filtered to remove the 20% of genes with the most extreme tissue-specificity values (i.e., the $n_{i,j}^*$ statistics). To test for pathways enriched in genes that are up-regulated in the cancer as compared to the associated normal tissue (panel a), filtering was performed to remove the most tissue-specific genes. To test for pathways enriched in genes that are down-regulated in the cancer as compared to the associated normal tissue (panel b), filtering was performed to retain tissue-specific genes. Similar Q-Q plots based on just a single cancer/normal pair can be found in Figs F and G in S1 Text.

## Use of normal tissue-specificity to improve cancer survival analysis

Normal tissue-specificity can also be used to improve the power of cancer survival analysis. As shown in Table 4, tissue-specific genes are more likely to be favorably prognostic for cancer

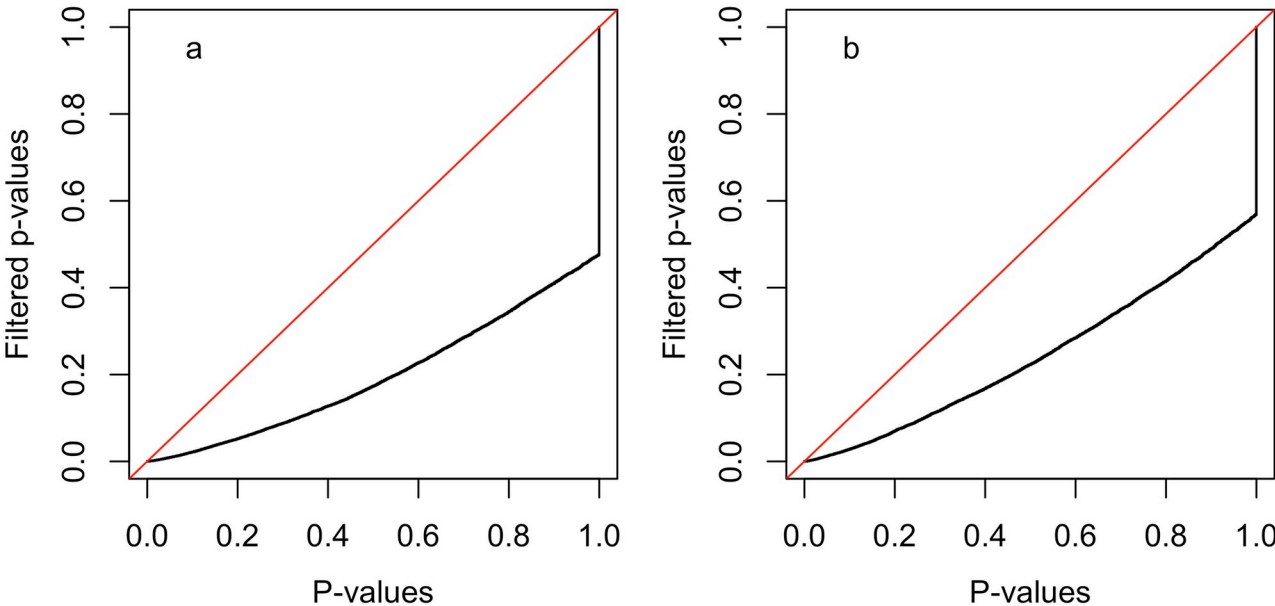

**Fig 5. Quantile-quantile plots illustrating the impact of tissue-specific gene filtering on gene set testing of MSigDB Hallmark pathways using cancer/normal relative expression ($r_{i,j}$).** Each plot contrasts the distribution of p-values (the black line) from tests for all 50 Hallmark pathways for each of the 21 cancer/normal pairs (total of 1050 tests) using all genes or genes filtered according to tissue-specificity. The red line has a slope of 1.0 and captures the distribution that would be expected if filtered and un-filtered p-values had the same distribution. In both panels, the x-axis reflects the p-value distribution when all genes are included and the y-axis reflects the p-value distribution when genes are filtered according to normal tissue-specificity ($n_{i,j}^*$). Panel a) contains the results from tests for enrichment of large $r_{i,j}$ statistics among the gene set members (i.e., are gene set members more likely to be up-regulated in cancer vs. the normal tissue?) and gene filtering removed the 20% of genes with the largest tissue-specificity values (i.e., kept genes that are less tissue-specific). Panel b) contains the results from tests for enrichment of small $r_{i,j}$ statistics among the gene set members (i.e., are gene set members more likely to be down-regulated in the cancer vs. the normal tissue?) and gene filtering removed the 20% of genes with the smallest tissue-specificity values (i.e., kept genes that are more tissue-specific). For both panels, the p-values for gene sets whose direction of enrichment is opposite the target direction were set to 1.0, which causes the vertical portion of the black line. As illustrated in the Q-Q plots, gene filtering in both cases improved gene set testing statistical power.

survival with non-tissue-specific genes more likely to be unfavorably prognostic. Unlike the association between tissue specificity and cancer/normal DE, the association with survival is only pronounced in a subset of the profiled cancer types. For these cancer types, an approach similar to that outlined above for the comparative analysis of cancer and normal tissue can be used to prioritize genes for survival analysis. Fig 6 illustrates the increase in survival analysis power that can be achieved by filtering genes according to normal tissue-specificity. Similar to Fig 5, this figure contains quantile-quantile (Q-Q) plots of p-values from survival analyses for all 21 profiled cancer types using either all genes or a subset of genes filtered according to normal tissue-specificity (the $n_{i,j}^*$ statistics). In this case, the p-values are generated via Kaplan-Meir (KM) tests of the association of gene expression with cancer survival following the approach of Ulhen et al. [18]. For the analysis of favorably prognostic genes (panel a), the p-values for all unfavorable genes were set to 1.0 and filtering was performed to remove genes where $n_{i,j}^* < log(0.8)$, i.e., genes down-regulated in the associated tissue. For the analysis of unfavorably prognostic genes (panel b), the p-values for all favorable genes were set to 1.0 and filtering was performed to remove genes where $n_{i,j}^* > log(1.2)$, i.e., genes up-regulated in the associated tissue. Similar Q-Q plots based on just a single cancer/normal pair can be found in Figs H and I in S1 Text.

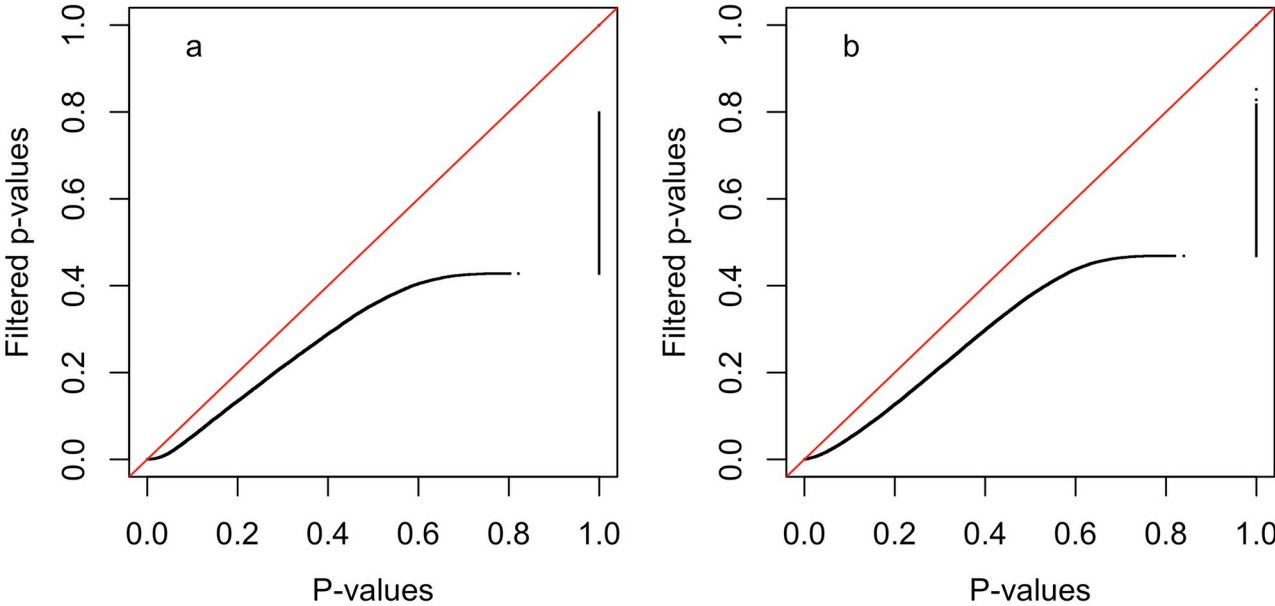

**Fig 6. Quantile-quantile plots illustrating impact of tissue-specific gene filtering on the distribution of p-values from Kaplan-Meir (KM) tests of the association of gene expression with cancer survival as computed using the approach of Ulhen et al. [18].** Each plot contrasts the distribution of p-values (the black line) for all genes for each of the 21 cancer types or for genes filtered according to tissue-specificity. The red line has a slope of 1.0 and captures the distribution that would be expected if filtered and un-filtered p-values had the same distribution. In both panels, the x-axis reflects the p-value distribution when all genes are included and the y-axis reflects the p-value distribution when genes are filtered according to normal tissue-specificity ($n_{i,j}^*$). Panel a) contains the results when only favorably prognostic genes are considered (i.e., genes where increased expression is associated with better survival; p-values for unfavorable genes were set to 1.0) and gene filtering removed genes where $n_{i,j}^* < log(0.8)$ (i.e., removed genes expressed at a lower level in the associated normal tissue than in the average tissue). Panel b) contains the results when only unfavorably prognostic genes are considered (i.e., genes where increased expression is associated with worse survival; p-values for favorable genes were set to 1.0) and gene filtering removed genes where $n_{i,j}^* > log(1.2)$ (i.e., removed genes expressed at a higher level in the associated normal tissue than in the average tissue). As illustrated in the Q-Q plots, gene filtering in both cases improved survival analysis statistical power.

## Limitations

There are a number of limitations associated with the data and analysis results presented in this paper that are important to note.

- The normal tissue paired with some of the cancer types (see Table 1) may not accurately reflect the tissue of origin or tissue microenvironment of the tumor, e.g., most melanoma samples in TCGA are metastatic lesions rather than primary tumors and most ovarian cancer tumors in TCGA are high-grade serious ovarian cancer which is thought to originate in the Fallopian tube rather than normal ovarian tissue [26, 27].

- The HPA and TCGA gene expression data used to compute the statistics described in Table 2 was measured on different individuals not matched for age, gender or clinical characteristics (this limitation is partly mitigated by the focus on mean expression across samples). Note that gender is largely consistent for the gender-specific cancers and associated tissues (breast, cervix, ovary, prostate, testis).

- The HPA RNA-seq data was measured on tissue samples from just three healthy individuals so the mean values can be expected to have much larger variances than the TCGA means. It should be noted that, despite the small number of samples, the mean gene expression values estimated from the HPA RNAs-seq data have been shown to provide accurate estimates of gene tissue-specificity as detailed in the original HPA paper [19].

- The rank correlation coefficients reported in Tables 1 and 4 are computed across all measured genes for each cancer type and normal tissue pair so have very large $n$ values and approximate p-values close to 0. Because the true $n$ values are much lower given the dependency structure between gene expression values, we only reported the correlation point estimates.

### Guidance on how to leverage normal tissue-specific associations for cancer transcriptomic analyses

To leverage the findings detailed in this paper for their own cancer transcriptomic analyses, researchers can adopt the following general approach:

- **Calculate the tissue-specific gene expression statistics $n^*_{i,j}$ for the normal tissue associated with the cancer type of interest.** For the 21 cancer types and 18 corresponding normal tissues investigated in this paper, these statistics can be found at the companion website (https://hrfrost.host.dartmouth.edu/CancerNormal/). For normal tissues not analyzed in this paper but supported by the HPA, it should be straightforward to compute the $n^*_{i,j}$ from the HPA RNA-seq data. For a normal tissue not supported by the HPA, the $n_{i,j}$ statistics can be computed from appropriate transcriptomic data; as long as the HPA normalization is used, $n^*_{i,j}$ statistics can be computed using the average $n_{i,j}$ statistics for the normal tissues profiled by the HPA.

- **For survival analysis using gene expression predictors:** Researchers can use the $n^*_{i,j}$ statistics to filter or weight the predictors to prioritize genes that are favorably and/or unfavorably prognostic. The approach detailed in the Use of normal tissue-specificity to improve cancer survival analysis Section above provides one example of how these statistics can be used for gene filtering. Researchers are encouraged to first check the association between prognostic ability and tissue-specificity (as captured in Table 4 and Fig E in S1 Text) to gauge the likelihood that filtering/weighting according to $n^*_{i,j}$ will be impactful for the cancer type of interest.

- **For the comparative analysis of gene expression in different cancers:** For the differential expression analysis between different cancer types or between primary and metastatic lesions of the same cancer, researchers can use the $n^*_{i,j}$ statistics to either prioritize genes that are differentially expressed (DE) between the corresponding normal tissues or prioritize genes that do not show normal tissue-specificity. Genes with large $n^*_{i,j}$ statistics for either normal tissue are likely to be DE between the corresponding cancers so filtering or weighting to prioritize large $n^*_{i,j}$ statistics will increase cancer DE power. However, as detailed in Section Using normal tissue gene activity to improve the comparative analysis of cancers, performing a cancer DE analysis that simply recapitulates the differences between the normal tissues is unlikely to be biologically informative. Instead, researchers can prioritize genes with $n^*_{i,j}$ statistics close to 0, i.e, genes that do not show significant up or down-regulation in the normal tissue, to highlight differences between the cancers that are independent of the normal tissues. The results from this type of analysis are shown in Table 5.

- **For the analysis of cancer/normal relative expression:** Because tissue-specific genes tend to be down-regulated in the corresponding cancer, researchers can use the $n^*_{i,j}$ statistics to filter or weight genes prior to the comparative analysis of normal and neoplastic tissue. The approach detailed in the Use of normal tissue-specificity to improve the comparative analysis

of normal and neoplastic tissue Section above provides one example of how these statistics can be used for gene filtering in this type of scenario. Similar to the survival analysis application, researchers are encouraged to first check the association between tissue-specificity and cancer/normal relative expression (as captured in Table 4 and Fig D in S1 Text) to gauge the likelihood that this type of filtering/weighting will be impactful for the cancer type of interest.

## Conclusion

The biology of human cancer is highly tissue-specific with the majority of cancer-related somatic alterations occurring in only a small number of tissue types and the functional impact of common and inherited mutations frequently exhibiting a tissue-specific functional impact. An important consequence of cancer tissue-specificity is that the biology of normal tissues holds important information regarding the molecular features of associated cancers, information that can be leveraged to improve the power and accuracy of cancer genomic analyses. To date, most research exploring the joint analysis of normal tissue and cancer genomic data has focused on the analysis of tumor and adjacent normal samples. The development of approaches that leverage the general characteristics of normal tissues for cancer analysis has only received limited attention with most investigations focusing on specific alterations within a single cancer type. To address this research gap and support use cases where adjacent normal tissue samples are unavailable, we have explored the genome-wide association between the transcriptomes of 21 solid human cancers profiled by TCGA and their associated normal tissues as profiled in healthy individuals by the HPA. Although the mean transcriptomic profiles of normal and cancerous tissue appear distinct, a strong association is revealed between each cancer and the corresponding normal tissue when gene expression data is transformed into tissue or cancer-specific values, i.e., the ratio of expression in one tissue or cancer relative to the mean in other tissues or cancers. As we have demonstrated through the analysis results presented in this paper, the strong association between cancer-specific and tissue-specific gene expression can be leveraged to significantly improve statistical power and biological interpretation for cancer survival analysis, cancer comparative analysis, and analysis of cancer/normal pairs.

## Supporting information

**S1 Text. Supporting methods and results.** The supporting information file includes supplemental methods and results (Tables A and B and Figures A-I, which are described below). **Table A. TCGA cancer type statistics.** The 21 analyzed TCGA cancer types, number of tumor samples with expression data, mean age, gender proportions, and corresponding HPA normal tissues. Note that gender proportions for some cancers may not add up to 1.0 if gender is unavailable for some samples. **Table B. Distances between each cancer type and the corresponding normal tissue in reduced principal component (PC) space.** The Euclidean distance was computed between the projections of each cancer type and the associated normal tissue in the space spanned by the all PC with non-zero variance of the mean gene expression matrix. The 'Cancer/normal relative distance' column contains the ratio of the distance between each cancer and normal tissue pair to the average distance between the cancer and all other cancers or normal tissues. The 'Cancer/normal-specific relative distance' column contains a similar distance ratio computed in the space spanned by the first two PCs of cancer-specific and normal tissue-specific mean expression values. The 'Distance ratio' column contains the ratio of the 'Cancer/normal-specific relative distance' to the 'Cancer/normal relative

distance'. **Fig A. Variance of the PCs of the mean gene expression matrix. Fig B. Projection of TCGA cancer types and associated HPA normal tissues onto principal components 3–8 of the mean expression values.** Principal components are computed from a matrix of mean gene expression values, i.e., a matrix containing all $c_{i,j}$ and $n_{i,j}$ statistics. a) Cancers are represented by blue points and are enclosed in the blue shaded region; normal tissues are represented by red points and are enclosed in the red shaded region. b) Each normal tissue and the associated cancer type(s) are enclosed in a separate shaded region. **Fig C. Projection of TCGA cancer types and associated HPA normal tissues onto principal components of the tissue/cancer-specific mean expression values.** Principal components are computed from a matrix of mean gene expression values, i.e., a matrix containing all $c_{i,j}$ and $n_{i,j}$ statistics. a) Cancers are represented by blue points and are enclosed in the blue shaded region; normal tissues are represented by red points and are enclosed in the red shaded region. b) Each normal tissue and the associated cancer type(s) are enclosed in a separate shaded region. **Fig D. Association between normal tissue-specificity ($n_{i,j}^{*}$ statistics) on the y-axis and cancer/normal relative expression ($r_{i,j}$ statistics) on the x-axis.** Each point in both plots represents a single gene and the red lines reflect the linear regression fit. **Fig E. Association between normal tissue-specificity ($n_{i,j}^{*}$ statistics) on the y-axis and cancer survival ($s_{i,j}$ statistics) on the x-axis.** Each point in both plots represents a single gene and the red lines reflect the linear regression fit. **Fig F. Quantile-quantile plots illustrating the impact of filtering tissue-specific genes on gene set testing of MSigDB Hallmark pathways using cancer/normal relative expression ($r_{i,j}$).** Each plot contrasts the distribution of p-values (the black line) from tests for enrichment of large $r_{i,j}$ statistics among the gene set members, i.e., are gene set members more likely to be up-regulated in cancer vs. the normal tissue? The x-axis reflects the p-value distribution when all genes are included and the y-axis reflects the p-value distribution when genes are filtered according to normal tissue-specificity ($n_{i,j}^{*}$). Gene filtering removed the 20% of genes with the largest tissue-specificity values (i.e., keeps genes that are less tissue-specific). **Fig G. Quantile-quantile plots illustrating the impact of filtering non-tissue-specific genes on gene set testing of MSigDB Hallmark pathways using cancer/normal relative expression ($r_{i,j}$).** Each plot contrasts the distribution of p-values (the black line) from tests for enrichment of small $r_{i,j}$ statistics among the gene set members, i.e., are gene set members more likely to be down-regulated in cancer vs. the normal tissue? The x-axis reflects the p-value distribution when all genes are included and the y-axis reflects the p-value distribution when genes are filtered according to normal tissue-specificity ($n_{i,j}^{*}$). Gene filtering removed the 20% of genes with the smallest tissue-specificity values (i.e., keeps genes that are more tissue-specific). **Fig H. Quantile-quantile plots illustrating the favorable impact of tissue-specific gene filtering on cancer survival analysis.** Each plot contrasts the distribution of p-values (the black line) capturing favorable prognostic status of all genes for genes filtered according to tissue-specificity. The x-axis reflects the p-value distribution when all genes are included and the y-axis reflects the p-value distribution when genes are filtered according to normal tissue-specificity ($n_{i,j}^{*}$). Gene filtering removed genes where $n_{i,j}^{*} < log(0.8)$ (i.e., removed genes expressed at a lower level in the associated normal tissue than in the average tissue). **Fig I. Quantile-quantile plots illustrating the unfavorable impact of tissue-specific gene filtering on cancer survival analysis.** Each plot contrasts the distribution of p-values (the black line) capturing unfavorable prognostic status of all genes for genes filtered according to tissue-specificity. The x-axis reflects the p-value distribution when all genes are included and the y-axis reflects the p-value distribution when genes are filtered according to normal tissue-specificity ($n_{i,j}^{*}$). Gene filtering removed genes where $n_{i,j}^{*} > log(1.2)$ (i.e., removed genes expressed at a higher level in the associated normal

tissue than in the average tissue).
(PDF)

## Acknowledgments

We would like to thank HPA group for providing access to the HPA.normal.FPKM.GDCpipeline.csv file.

## Author Contributions

**Conceptualization:** H. Robert Frost.

**Data curation:** H. Robert Frost.

**Formal analysis:** H. Robert Frost.

**Funding acquisition:** H. Robert Frost.

**Investigation:** H. Robert Frost.

**Methodology:** H. Robert Frost.

**Project administration:** H. Robert Frost.

**Resources:** H. Robert Frost.

**Software:** H. Robert Frost.

**Supervision:** H. Robert Frost.

**Validation:** H. Robert Frost.

**Visualization:** H. Robert Frost.

**Writing – original draft:** H. Robert Frost.

**Writing – review & editing:** H. Robert Frost.

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
