## [Decision Letter · Decision Letter 0]

5 Apr 2021

Dear Dr. Frost,

Thank you very much for submitting your manuscript "Analyzing cancer gene expression data through the lens of normal tissue-specificity" for consideration at PLOS Computational Biology.

As with all papers reviewed by the journal, your manuscript was reviewed by members of the editorial board and by several independent reviewers. In light of the reviews (below this email), we would like to invite the resubmission of a significantly-revised version that takes into account the reviewers' comments.

I think this study has potential. In the literature, only a handful of studies have paid sufficient attention to gene expression data of normal tissues. The paper is overall well written (although reviewers have noticed a few typos and small mistakes). The analysis is mostly rigorous, and the findings can be potentially informative.

Limited literature is cited. This can be improved to better position this study.

A possibly minor technical question: when constructing the PCs, are all genes used? Most genes are not cancer related, and including all (which contain a huge amount of noises) can be problematic.

Why the first two PCs? Statistically and biologically, there is no reason why, for example, the second PC is more important than the third one.

I understand the importance of looking into gene expressions of normal tissues. However, it is still unclear to me how I can potentially use findings in this article in translational/clinical studies. This aspect needs to be significantly strengthened.

Some of the descriptions are not sufficiently rigorous. For example, “very similar”: is there an objective quantification? Is the difference statistically significant?

TCGA has more than 21 cancer types. Why select those 21?

We cannot make any decision about publication until we have seen the revised manuscript and your response to the reviewers' comments. Your revised manuscript is also likely to be sent to reviewers for further evaluation.

Sincerely,

Shuangge Ma

Guest Editor

PLOS Computational Biology

Florian Markowetz

Deputy Editor

PLOS Computational Biology

I think this study has potential. In the literature, only a handful of studies have paid sufficient attention to gene expression data of normal tissues. The paper is overall well written (although reviewers have noticed a few typos and small mistakes). The analysis is mostly rigorous, and the findings can be potentially informative.

Limited literature is cited. This can be improved to better position this study.

A possibly minor technical question: when constructing the PCs, are all genes used? Most genes are not cancer related, and including all (which contain a huge amount of noises) can be problematic.

Why the first two PCs? Statistically and biologically, there is no reason why, for example, the second PC is more important than the third one.

I understand the importance of looking into gene expressions of normal tissues. However, it is still unclear to me how I can potentially use findings in this article in translational/clinical studies. This aspect needs to be significantly strengthened.

Some of the descriptions are not sufficiently rigorous. For example, “very similar”: is there an objective quantification? Is the difference statistically significant?

TCGA has more than 21 cancer types. Why select those 21?

Reviewer's Responses to Questions

**Comments to the Authors:**

Reviewer #1: Highly interesting work on how best to disentangle cancer-specific processes from tissue-specific ones. With a few revisions to ensure clarity, this paper will be a valuable contribution to the field of cancer transcriptomics.

Major revisions:

-Major elements of the paper rely on the comparison between cancer and corresponding normal tissues, but it is not discussed in the paper how these pairings were made. Ostensibly the information is based on which tissues were available from HPA, but this needs to be made explicit in the body of the text. Selecting an appropriate normal tissue to compare to is not a trivial problem for many cancer types. For instance, this paper pairs ovarian cancer (OV) with normal ovarian tissue. The overwhelming majority of ovarian cancer cases in TCGA are high-grade serious ovarian cancer, which is believed to frequently arise in the Fallopian tube and more closely resembles normal Fallopian tissue than normal ovarian tissue. The challenge of pairing cancers and normal tissues is highlighted in Table 2, where several cancer types were more correlated with another tissue type than the one selected. It is essential that the pairing decisions are described and justified.

-While there’s an obvious reason to use only the first two principal components when plotting, the distance calculations shown in Table 3 are feasible on a range of principal components and likely to be highly affected by number of PCs used. The paper needs some sort of explanation or justification of the selection of 2 principal components, including percentage of variance explained by PCs 1 and 2 and ideally an elbow plot of percent variance explained by a range of PCs.

Minor revisions:

-The manuscript is not organized according to the standards outlined in the PLOS Computational Biology submission guidelines. Reorganizing into the standard sections (Introduction, Results, Discussion, Materials and Methods) could be done without too much alteration of the text itself.

-The headings (ie the bolded first sentence) for Figure 1 and Figure 2 are identical; the header should indicate what is different between these two figures.

-The right panel of Figures 1 and 2 is very difficult to interpret, with many overlapping shaded regions. For visual clarity, using lines to connect single cancer-normal tissue pairs and only using shaded ovals for the multi-associations would communicate the results of the figure much more clearly.

-Normal tissue data from HPA, the file “HPA.normal.FPKM.GDCpipeline.csv” as described in the supplemental methods, is not clearly accessible. A link to this data should be provided.

Reviewer #2: This article has studied the normal tissue data for the analysis of cancer genomics. Advanced from the existing studies that usually focused on the paired analysis of tumor and adjacent normal samples, this study has explored the genome-wide association between the transcriptomes of 21 TCGA cancer types and their associated normal tissues as profiled in healthy individuals from the Human Protein Atlas. Some interesting findings have been observed, where there is a strong association between tissue-specific and cancer-specific expressions, and this association can be leveraged to improve the prognostic modeling of cancer, the comparative analysis of different cancer types, and the analysis of cancer and normal tissue pairs. Overall, this article is biologically interesting. However, more detailed discussions would be added to improve the presentation. Specifically, I have a number of concerns regarding the paper.

1. Since the results presented in the paper are based on data from TCGA and HPA, it is suggested to provide more detailed descriptions on these data. For example, for each cancer type or normal tissue, it is better to provide the sample size and dimension of genes. Is any preprocessing conducted on the original downloaded data, such as the matching of the genes in different cancer types or normal tissues, prescreening to reduce the number of genes, or standardization on the gene expression measurements?

2. PCA is conducted on the matrix consisting of all c_{ij} and n_{ij} statistics, and also that consisting of all c*_{ij} and n*_{ij} statistics, where the sample size is 38. What is the total number of c_{ij} and n_{ij} statistics? The PCA approach may be infeasible, as the number of genes is usually much larger than the sample size (38).

3. Would you please provide more detailed discussions on the definitions of c*_{ij} and n*_{ij} statistics, which are referred to as tissue specific and cancer-specific gene expressions. For example, why log2 transformation is conducted for c*_{ij} and n*_{ij} statistics, but not for c_{ij} and n_{ij} statistics. Why c*_{ij} and n*_{ij} can be referred to as tissue specific and cancer-specific gene expressions? They are in a sense the simple scaled values of c_{ij} and n_{ij} statistics.

4. In section “Association between normal tissue-specificity, cancer/normal differential expression and cancer survival”, it is demonstrated that “an increase in expression of normal tissue specific genes is associated with improved survival in cancer. Genes that

are not tissue-specific have the inverse association, i.e., an increase in expression is associated with worse survival. Genes that are down-regulated in a tissue relative to other tissues tend to have no survival association”. Would you provide more detailed discussions on these statements? For example, how can we distinguish genes that are normal tissue specific or not? Why the results in Table 4 can support these statements?

5. Figure 4. “Cell coloring reflects the Pearson correlation between the fold-change in gene expression between each pair of cancers and the corresponding pair of normal tissues” is confusing. It is better to provide more details on the calculation of the corresponding Pearson correlation.

6. In section “Using normal tissue gene activity to improve the comparative analysis of cancers”, the results support that “when genes exhibiting significant normal tissue DE are removed, the comparative cancer results are distinct from the normal tissue results.” However, what is the improvement of using normal tissue gene activity for the comparative analysis of cancers.

Reviewer #3: • In Table 2, what are the sample size, and the distribution of age and gender for each of cancers and HPA tissues? Largely different proportions of those factors in these two groups (cancer and HPA tissue) can lead to very biased or misleading results. The author only reported the Spearman’s correlation without the statistical inference (eg. p-value). One of the author’s key claims in this manuscript is that “majority of the profiled cancers are mostly strongly correlated with their corresponding normal tissue.” However, if I understood Table 2 correctly, this seems to be true for only 12 out of 21 cases, and thus the author’s claim is not strongly supported.

• For certain gender dominant cancers, eg. prostate and breast, it is quite natural to think that they are more correlated with their corresponding normal tissue, compared to other non-corresponding tissues since all data consists of the same gender, which seemed to be also supported by the results from Table 2. However, gender specificity was ignored in this manuscript. This should not be the issue for the paired tumor/normal analysis, but I think in this aggregated approach, at least important biological factors such as gender should be controlled or matched to infer the role of normal tissues more accurately.

• The author advocated the new measures of c* and n*, which is basically simple log transformation of the original cancer and normal tissue specific gene expression, respectively, while subtracting their batch effects. The author argued that after using the new measures, the PCA results indicate that normal tissue and cancer are no longer separated. However, I doubt about the logical reasoning of how the author came to this conclusion. First of all, the usage of different scales of x (and y ) axes in the figures 1 and 2 is misleading. Fixing the same scale in both figures, the difference may not be large enough as the author originally considered. Most of all, there is no mathematical or heuristic justification of the new measures of c* and n* regarding why they are better measures than c and n.

Overall, although the motivation of the manuscript is interesting, the added value of this paper to the research community is not significant. Moreover, the manuscript has much room for improvement in several ways. More importantly, rudimentary typos (eg. BRAC 1/BRAC2 rather than BRCA1/BRCA2, reverse quotations marks at the start of a quotation in many places) could have been avoided by carefully proofreading before submission.

**Have the authors made all data and (if applicable) computational code underlying the findings in their manuscript fully available?**

Reviewer #3: **No: **

PLOS authors have the option to publish the peer review history of their article (what does this mean?). If published, this will include your full peer review and any attached files.

Reviewer #1: No

Reviewer #2: No

Reviewer #3: No

**Have all data underlying the figures and results presented in the manuscript been provided?**

Reviewer #1: **No: **Normal tissue data from HPA, the file “HPA.normal.FPKM.GDCpipeline.csv” as described in the supplemental methods, is not clearly accessible. A link to this data should be provided.

Reviewer #2: Yes
---

## [Decision Letter · Decision Letter 1]

12 May 2021

Dear Dr. Frost,

Thank you very much for submitting your manuscript "Analyzing cancer gene expression data through the lens of normal tissue-specificity" for consideration at PLOS Computational Biology. As with all papers reviewed by the journal, your manuscript was reviewed by members of the editorial board and by several independent reviewers. The reviewers appreciated the attention to an important topic. Based on the reviews, we are likely to accept this manuscript for publication, providing that you modify the manuscript according to the review recommendations.

The revision is largely satisfactory. One reviewer still has some very minor lingering concerns -- please address them properly.

Otherwise, congratulations on a job nicely done.

Sincerely,

Shuangge Ma

Guest Editor

PLOS Computational Biology

Florian Markowetz

Deputy Editor

PLOS Computational Biology

[LINK]

The revision is largely satisfactory. One reviewer still has some very minor lingering concerns -- please address them properly.

Otherwise, congratulations on a job nicely done.

Reviewer's Responses to Questions

**Comments to the Authors:**

Reviewer #1: The paper is substantially improved compared to the first submission. The previous concerns about justifying number of principal components used have been sufficiently addressed, and the inclusion of a limitations section and guidance to researchers on how to perform similar analyses are beneficial.

Minor revisions:

- There is still a paucity of citations in the paper. Several statements in the paper ought to be combined with a source, for instance:

- Introduction lines 45-48, “While many researchers have investigated the association between normal tissue biology and cancer development in the context of specific cancer types and the associated cancer drivers (e.g., the association between estrogen sensitive tissues, BRAC1/BRAC2 mutations and cancer development)” The relevant work, or at least one or two representative papers from this work, should be cited.

- Limitations section lines 289-291, re: how high-grade serous ovarian tumors are thought to originate in the Fallopian tube.

- As Reviewer 3 noted, the genes BRCA1/BRCA2 are incorrectly referred to as BRAC1/BRAC2, and this doesn’t appear to have been changed in the resubmission.

- The first paragraph of the section “Use of normal tissue-specificity to improve cancer survival analysis”, specifically lines 264-268, is slightly jumbled post-revision. Consider revising for clarity.

Reviewer #2: All my comments have been addressed. I have no further comment.

Reviewer #3: The authors have addressed all the issues raised

**Have the authors made all data and (if applicable) computational code underlying the findings in their manuscript fully available?**

Reviewer #1: Yes

Reviewer #2: Yes

Reviewer #3: None

PLOS authors have the option to publish the peer review history of their article (what does this mean?). If published, this will include your full peer review and any attached files.

Reviewer #1: No

Reviewer #2: No

Reviewer #3: No

Figure Files:

Data Requirements:

Reproducibility:

References:

---

## [Editor Report · Decision Letter 2]

15 May 2021

Dear Dr. Frost,

We are pleased to inform you that your manuscript 'Analyzing cancer gene expression data through the lens of normal tissue-specificity' has been provisionally accepted for publication in PLOS Computational Biology.

Best regards,

Shuangge Ma

Guest Editor

PLOS Computational Biology

Florian Markowetz

Deputy Editor

PLOS Computational Biology

---

## [Editor Report · Acceptance letter]

10 Jun 2021

PCOMPBIOL-D-21-00181R2 

Analyzing cancer gene expression data through the lens of normal tissue-specificity

Dear Dr Frost,

I am pleased to inform you that your manuscript has been formally accepted for publication in PLOS Computational Biology. Your manuscript is now with our production department and you will be notified of the publication date in due course.

With kind regards,

Katalin Szabo
